# Polyphenols Epigallocatechin Gallate and Resveratrol, and Polyphenol-Functionalized Nanoparticles Prevent Enterovirus Infection through Clustering and Stabilization of the Viruses

**DOI:** 10.3390/pharmaceutics13081182

**Published:** 2021-07-31

**Authors:** Dhanik Reshamwala, Sailee Shroff, Olivier Sheik Amamuddy, Valentino Laquintana, Nunzio Denora, Antonella Zacheo, Vili Lampinen, Vesa P. Hytonen, Özlem Tastan Bishop, Silke Krol, Varpu Marjomäki

**Affiliations:** 1Department of Biological and Environmental Science/Nanoscience Center, University of Jyväskylä, 40014 Jyväskylä, Finland; dhanik.d.reshamwala@jyu.fi (D.R.); sailee.s.shroff@jyu.fi (S.S.); 2Research Unit in Bioinformatics (RUBi), Department of Biochemistry and Microbiology, Rhodes University, Makhanda 6140, South Africa; oliserand@gmail.com (O.S.A.); o.tastanbishop@ru.ac.za (Ö.T.B.); 3Department of Pharmacy–Pharmaceutical Sciences, University of Bari “Aldo Morro”, 70125 Bari, Italy; valentino.laquintana@uniba.it (V.L.); nunzio.denora@uniba.it (N.D.); 4Laboratory for Nanotechnology, IRCCS Istituto Tumori “Giovanni Paolo II”, 70124 Bari, Italy; antonellazacheo@gmail.com; 5Faculty of Medicine and Health Technology, Tampere University, 33520 Tampere, Finland; vili.lampinen@tuni.fi (V.L.); vesa.hytonen@tuni.fi (V.P.H.); 6Fimlab Laboratories, 33520 Tampere, Finland; 7Laboratory for Personalized Medicine, National Institute of Gastroenterology, “S. de Bellis” Research Hospital, 70013 Castellana Grotte, Italy; silke.krol@aol.com

**Keywords:** polyphenols, functionalized gold nanoparticles, antiviral efficacy, enteroviruses, stabilization

## Abstract

To efficiently lower virus infectivity and combat virus epidemics or pandemics, it is important to discover broadly acting antivirals. Here, we investigated two naturally occurring polyphenols, Epigallocatechin gallate (EGCG) and Resveratrol (RES), and polyphenol-functionalized nanoparticles for their antiviral efficacy. Concentrations in the low micromolar range permanently inhibited the infectivity of high doses of enteroviruses (10^7^ PFU/mL). Sucrose gradient separation of radiolabeled viruses, dynamic light scattering, transmission electron microscopic imaging and an in-house developed real-time fluorescence assay revealed that polyphenols prevented infection mainly through clustering of the virions into very stable assemblies. Clustering and stabilization were not compromised even in dilute virus solutions or after diluting the polyphenols-clustered virions by 50-fold. In addition, the polyphenols lowered virus binding on cells. In silico docking experiments of these molecules against 2- and 3-fold symmetry axes of the capsid, using an algorithm developed for this study, discovered five binding sites for polyphenols, out of which three were novel binding sites. Our results altogether suggest that polyphenols exert their antiviral effect through binding to multiple sites on the virion surface, leading to aggregation of the virions and preventing RNA release and reducing cell surface binding.

## 1. Introduction

The gold standard in viral disease management is prevention, which is vaccination. Pandemics with new viruses remind us that the development and approval of vaccines takes time. Moreover, vaccines are not 100% efficient, and especially elderly people and immunocompromised individuals may not generate sufficient protection against infectious agents. Therefore, powerful broadly acting antivirals with a more general inhibitory mechanism of action are required.

Non-enveloped enteroviruses are very stable and stay infective on surfaces and in the environment for long periods. Their entry is mostly via the gastro-intestinal tract, but also through the upper respiratory system. They cause a high number of acute infections such as flu, aseptic meningitis and myocarditis, but also contribute to chronic diseases such as dilated cardiomyopathy, asthma or type I diabetes [1]. For enteroviruses, no approved antivirals exist, and there are vaccines available only against poliovirus and EV71 [1]. Therefore, it is necessary to reduce the viral load on the surfaces and in the environment and prevent infection in the primary infection sites. An optimal solution to tackle virus infection would be a drug that directly kills virus infectivity and additionally prevents the virus from entering cells.

Nature provides a large variety of potential antimicrobials and antivirals. Those may be of endophyte origin, secreted by the symbiotic fungi or bacteria living in plants [2]. One example is Resveratrol (3,5,4′-trihydroxy-trans-stilbene), which is a polyphenol, non-flavonoid compound and is produced by plants in response to cellular damage or pathogen attacks [3]. Resveratrol (RES) is found in strongly pigmented vegetables and fruits such as the skin of grapes, blueberries and peanuts, and it has strong bioactive potential [3,4,5]. Like RES, flavonoids are polyphenols found as secondary metabolites in vegetables, fruits, nuts, red wine and tea. They share great similarity in structure to RES and great promise to be used as antivirals [6,7,8].

In the present study, EGCG and RES were tested for their antiviral efficacy, and the mechanisms of action was elucidated against the stable non-enveloped enterovirus B group viruses. While RES is considered completely non-toxic [9], high amounts of free EGCG show in vitro hepatotoxic effects, and in particular in cases in vivo [10,11]. Therefore, here, free polyphenols as well as polyphenols functionalized onto gold nanoparticles (AuNPs) were studied for their antiviral efficiency, as binding polyphenols to AuNPs can reduce toxicity. The effect of reduced toxicity by nanoparticle binding has been shown previously for molecules with a known hepatotoxic profile [12]. AuNPs were synthesized by green chemistry using the polyphenols as redox agents to further reduce the toxic components in the preparations. AuNPs synthesized using green chemistry have been previously reported to act as antivirals [13].

In this work, we show that both RES and EGCG have a direct and long-lived antiviral effect through binding to multiple sites on the virus particles, preventing the virus from opening, reducing the binding to the cell surface and inducing clustering of the virions.

## 2. Materials and Methods

### 2.1. Cells

Human alveolar basal epithelial adenocarcinoma (A549) cells and Green Monkey Kidney (GMK) cells were obtained from American type culture collection (ATCC, Manassas, VA, USA). The A549 and GMK cell lines were propagated in Dulbecco’s Modified Eagle Medium (DMEM) (Gibco, Paisley, UK) and Eagle’s Minimum Essential Medium (MEM) (Gibco, Paisley, UK) respectively, supplemented with 10% Fetal Bovine Serum (FBS, Gibco, Paisley, UK), 1% L-GlutaMAX (Gibco, Paisley, UK) and 1% antibiotics (penicillin/streptomycin) (Gibco, Paisley, UK) in a humidified 5% CO_2_ incubator at 37 °C.

### 2.2. Viruses

Coxsackievirus B3 (CVB3; Nancy strain), Coxsackie virus B1 (CVB1, CONN 5 strain) and Coxsackievirus A9 strain (CVA9; Griggs strain), obtained from ATCC, were produced and purified as described before [14,15,16], with the only exception of adding 0.1% (*v*/*v*) TWEEN^®^ 80 (Sigma-Aldrich, Steinheim, Germany) during the freeze–thaw cycle.

### 2.3. Polyphenols

Epigallocatechin gallate (EGCG) and Resveratrol (RES) were purchased from Sigma-Aldrich (Munich, Germany). EGCG was dissolved in water and Resveratrol in 0.02 M NaOH. Stock concentrations of EGCG and RES were 6.5 mM (3 mg/mL) and 8.8 mM (2 mg/mL) respectively, and were stored at −20 °C.

### 2.4. Nanoparticle Preparations

The green chemistry approach using the polyphenolic compounds as redox agents was chosen to reduce the number of potentially toxic molecules present on the surface of the gold nanoparticles (AuNPs). These polyphenols are supposed to form a stable and stabilizing layer on the AuNPs, as it was observed for citrate AuNPs shown by Mandal et al. [17]. For the preparation of RES-coated gold nanoparticles (RES-AuNPs), we used a modified protocol by Sanna et al. [18]. In brief, 20.8 mg of HAuCl_4_·3 H_2_O (Sigma-Aldrich, Milan Italy) was dissolved in 12 mL of bi-distilled water on ice. Next, an ice-cold light-yellow solution of 3.0 mg of RES in 1.5 mL of 0.02 M NaOH solution was quickly added under vortexing. Under continuous vortexing, the temperature of the solution was slowly raised to room temperature (RT). Immediately after RES addition, the solution became brownish, after 10 min reddish, and after vortexing at 500 rpm for 1 h, the color turned into wine red. Unbound RES was removed by a series of washing (first wash: 0.02 M NaOH; second wash: H_2_O) and centrifugation steps (2 × 14,000 rpm, 20 min at 4 °C to remove NaOH; 1 × 12,000 rpm, 20 min at 4 °C to remove H_2_O), and the pellet was finally resuspended in water. The centrifugation was carried out by a Microfuge 40R with a Fiberlite F21-48 x2 Fixed Angle Rotor, 45°, R_max_ 97 mm (ThermoFisher, Rome, Italy). The final gold concentration was 5.3 mg/mL, as determined by ICP-MS. The concentration of RES in the stock solution was calculated (Appendix A) to be 34.4 μM.

For the preparation of EGCG-AuNPs, 5.9 mg of EGCG was dissolved in 15 mL of bi-distilled water and cooled on ice. An ice-cold solution of 12.9 mg of HAuCl_4_·3 H_2_O in 2 mL of H_2_O was added quickly and mixed by shaking on ice for another 10 min. After the color changed from bright yellow to red within 1 min, the temperature of the solution was slowly increased to RT for 2 h under continuous vortexing at 500 rpm. The excess EGCG was removed by a series of washing and centrifugation steps (2× centrifugation at 14,000 rpm for 20 min at 4 °C addition of ddH_2_O, 1× centrifugation at 12,000 rpm for 20 min at 4 °C). The deep red pellet was resuspended in water. The final gold concentration was 1.64 mg/mL, as determined by ICP-MS. The concentration of EGCG in the stock solution was calculated (Appendix A) to be 5.61 μM.

#### Quantitative Analysis of EGCG and RES Bound to the AuNP Surface

The amount of EGCG and RES present on the surface of the gold nanoparticles was estimated indirectly by the substitution with l-cysteine. The thiol group of the cysteine binds stronger to the gold than the aromatic ring system of polyphenols, and therefore will replace it. The colorimetric method finally determines the remaining free thiol groups of the l-cysteine, assuming a complete substitution of the polyphenols by cysteine [19]. The residual-free thiol groups of l-cysteine in solution after polyphenol exchange on the nanoparticle surface were quantified photometrically using Ellman’s reagent (5,5′-dithio-bis (2-nitrobenzoic acid)). The amount of polyphenols was then calculated considering a 1:1 ratio of replacement (1 cysteine replaces 1 polyphenol).

For its determination, a fresh stock solution of 1 mg of cysteine in 1 mL of 0.5 M phosphate buffer, pH 8.0, was diluted to a final concentration in l-cysteine of 102 mM. 100 μL of this solution was mixed with different volumes of EGCG-AuNPs or RES-AuNPs stock solution (ranging from 10 to 100 μL) and 0.5 M phosphate buffer solution at pH 8.0 to reach a final solution volume of 500 μL. This was mixed with 500 mL of Ellman’s reagent solution (3 mg dissolved in 10 mL of 0.5 M phosphate buffer, pH 8.0). The resulting mixture was incubated in the dark and at room temperature for 2 h. After this time, the samples were centrifuged at 13,000 rpm for 5 min (Mikro 22 R, Hettich Zentrifugen, Germany) and 100 µL of each sample was transferred into a microplate reader (PerkinElmer VICTOR X3 Multilabel) to measure absorbance at 450 nm. The amount of thiol groups was estimated from a standard curve of l-cysteine.

### 2.5. Dynamic Light Scattering Measurements

For dynamic light scattering (DLS) measurements, CVA9 virus samples were diluted in PBS from a stock solution (5 × 10^10^ PFU/mL) to 2 × 10^9^, 2 × 10^8^, 5 × 10^7^ and 2 × 10^7^ PFU/mL. EGCG was added to each of these dilutions such that its final concentration was 6.5 μM. Experimental virus controls with the same dilutions but without the EGCG were also prepared. Both the control virus and EGCG-treated virus were incubated for 1 h at 37 °C. Post-incubation, the virus/virus cluster size distribution was measured by DLS at 25 °C with the Zetasizer Nano ZS instrument (Malvern Instruments, Malvern, UK). The attenuator was set by the system according to scattering intensity of the sample and set measurement position at 3 mm. As controls, we measured both EGCG and PBS buffer alone. As expected, no particles were observed in the controls.

### 2.6. Antiviral Activity Assay

The antiviral activity of the polyphenols against species B enteroviruses was determined using a cytopathic effect (CPE) inhibition assay, modified from Schmidtke et al. [20]. Briefly, A549 cells were cultured for 24 h at 37 °C in 96-well flat-bottomed microtiter plate (Sarstedt, Numbrecht, Germany) at a density of 12,000 cells/well in 100 µL of DMEM supplemented with 10% FBS, 1% GlutaMAX and 1% penicillin/streptomycin antibiotics. The following day, each of the viruses (2 × 10^7^ PFU/mL for CVB3 and CVB1 and 2 × 10^8^ PFU/mL for CVA9) were pre-incubated with the compounds for 1 h at 37 °C. The virus–compound mixture was further diluted 10 times with DMEM to obtain a final MOI of 10 for CVB3 and CVB1 and 100 for CVA9. This mix was then added to cells for 24 h in a humidified 5% CO_2_ incubator at 37 °C. A virus without the compound was used as a positive control and a mock infection without the virus and compound as a negative control for the experiments. The development of a cytopathic effect (CPE) was monitored using light microscopy. The next day, the cells were fixed and stained for 10 min with CPE dye (0.03% crystal violet, 2% ethanol and 36.5% formaldehyde) after washing twice with PBS. After two washes with water, the stained viable cells were lysed using a lysis buffer (0.8979 g of sodium citrate and 1N HCl in 47.5% ethanol) to elute the crystal violet. Then, the absorbance was measured spectrophotometrically at 570 nm using the VICTOR^TM^ X4 multilabel reader (PerkinElmer, Turku, Finland). This assay was performed twice independently for each of the virus–compound concentrations.

### 2.7. Cytotixicity Assay

Cytotoxicity studies were performed using the CPE inhibition assay by adding the compounds (without any virus) to A549 cells. Mock infection was used as a negative control. This assay was performed twice independently.

### 2.8. Time of Addition Studies

Time of addition studies were performed using the CPE inhibition assay. Briefly, A549 cells were cultured for 24 h in 96-well flat-bottomed microtiter plate at a density of 12,000 cells/well in 100 µL of DMEM supplemented with 10% FBS, 1% GlutaMAX and 1% penicillin/streptomycin antibiotics. On the subsequent day, cells were infected with CVB3 (MOI 10) for 1 h at 37 °C. Afterwards, excess virus was removed by repeated washing. Then, fresh media (supplemented with 10% FBS, 1% GlutaMAX and 1% penicillin/streptomycin) containing EGCG (65 μM) and RES (880 μM) were added, and cells were incubated for 24 h. Virus control (without the compounds), compound control (without virus) and mock infection were used as controls for the studies. This assay was performed twice independently for each of the virus–compound concentrations, with 5 technical replicates in each experiment.

### 2.9. Radioactive Labeling of CVA9

The radioactive virus was produced in confluent monolayers of GMK cells, which were washed and incubated with PBS for 15 min at 37 °C and infected with CVA9 (Griggs strain; ATCC) diluted in low-methionine-cysteine medium (MP Biomedicals, Illkrich, France) supplemented with 1% FBS and 1% Glutamax. After 3 h of infection, 40 μCi/mL of radioactive [^35^S] methionine/cysteine (PerkinElmer, Boston, MA, USA) diluted in low-methionine-cysteine medium supplemented with 1% FBS was added to the cells, and virus replication was allowed to proceed for 24 h at 37 °C. After three freeze–thaw cycles, the supernatant was cleared by pelleting at 2500× *g* for 10 min. TWEEN^®^ 80 (0.1% *v/v*) was added to the pellet and incubated on ice for 30 min. Post-incubation, centrifugation was performed for 10 min with 4000× *g* at 4 °C. The collected supernatant was added on a 2 mL sucrose cushion (40%). The cushions were ultracentrifuged with an SW-41 rotor at 151,263× *g* for 2.5 h (4 °C). All the liquid up until the cushion and first 500 μL fraction from the top were discarded, and the next 3 × 500 μL fractions were collected, which were pelleted with a 70Ti rotor at 90,140× *g* for 2 h at 4 °C. The concentrated virus was then layered on top of the 10 mL sucrose gradient (5–20%) and ultracentrifuged with a SW-41 rotor at 151,263× *g* for 2 h (4 °C). Fractions (500 µL) were collected from the top to the bottom of the cushion and small samples from each fraction were mixed with a scintillation cocktail UltimaGold^TM^ (PerkinElmer, Waltham, MA, USA) for counting the radioactivity (counts per minute) using a Tri-Carb^®^ 2910TR liquid scintillation analyzer (PerkinElmer, Downers Grove, IL, USA). The 160S-containing fractions were pooled and used for the experiments (7000 CPM/μL; 1.4 × 10^9^ PFU/mL).

### 2.10. Gradient Assay

Metabolically labeled CVA9 (70,000 CPM corresponding to 1.4 × 10^8^ PFU/mL) was pre-treated with EGCG (6.5 μM) and RES (1.8 mM) for 1 h at 37 °C. A virus control (without the compounds but the same treatment) and fresh virus (without the polyphenols and without treatment) were used as positive controls for the assay. The test and control samples were further diluted 10 times using a buffer, PBS-MgCl_2_ (PBS containing 2 mM MgCl_2_). The samples were then layered over 10 mL sucrose (5–20%) gradients and ultracentrifuged with a SW-41 rotor at 151,263× *g* (average g value) for 2 h (4 °C) using the SW-41 rotor. Gradients were fractionated from top to bottom (500 μL fractions) and mixed with a scintillation cocktail for counting the radioactivity using a PerkinElmer Tri-Carb^®^ 2910TR liquid scintillation analyzer.

### 2.11. Real-Time Fluorescence Uncoating Assay

The real-time fluorescence uncoating assay has been previously described for measuring uncoating of enteroviruses at 37 °C [16,21]. The assay is based on the fluorescence emitted by SYBR Green II (SGII) when bound to viral genomic RNA. The experiment was carried out in a 96-well plate with a total reaction volume of 100 μL. Each well contained 0.5 μg of Coxsackievirus B3, 10× SGII (Life Technologies, Eugene, OR, USA) diluted in double-distilled water (ddH_2_O), polyphenol-functionalized nanoparticles EGCG-AuNPs (Au concentration 16.4 μg/mL, ligand concentration 5.61 × 10^−2^ μM) and RES-AuNPs (Au concentration 53 μg/mL, ligand concentration 3.44 × 10^−1^ μM) in storage buffer PBS-MgCl_2_ and opening buffer (20 mM NaCl, 6 mM KH_2_PO_4_, 12 mM K_2_HPO_4_ and 0.01% faf-BSA). We had two experimental controls: one was a virus in the storage buffer and the other contained a virus in the opening buffer. The samples were prepared on ice to prevent any effect of temperature on uncoating. Additionally, we had controls to exclude fluorescence quenching/enhancement from other molecules in the reaction mix. RNAse at a final concentration of 10 μg/mL was used to distinguish between the fluorescence originating from porous intermediate virions (allowing entry of SGII inside) and the RNA in solution. The fluorescence was recorded every 15 min for an hour at 37 °C using the PerkinElmer Multilabel Reader Victor X4 installed with a F485 lamp filter, F535 emission filter and a counting time of 1 s.

### 2.12. Particle Stability Thermal Release Assay (PaSTRy)

The PaSTRy assay was performed as described before [22]. The experiment was carried out in thin-walled PCR plates (Agilent, Amstelveen, Netherlands). A reaction mixture containing 0.5 μg of Coxsackievirus B3 and EGCG (65 μM) in storage buffer or opening buffer was preincubated for 1 h at 37 °C and 5% CO_2_. Post-incubation, 10× SGII (Invitrogen) in ddH_2_O was added to the reaction mix, and a final volume of 50 μL was aliquoted into the thin-walled PCR plates. The thermal cycler (BioRad C100, Helsinki, Finland) recorded the fluorescence in quadruple from 20 to 90 °C with 0.5 °C intervals. The fluorescence data output was extracted from the BioRad CFX manager (2.1 software, accessed on 10 October 2020) and processed in Microsoft Excel (2016). The melt curve was obtained by plotting the relative fluorescence units (RFU) values versus temperature. The melting temperature (Tm) was determined from the melt peak, which was plotted using the derivative of the RFU as a function of temperature (d(RFU)/dT).

### 2.13. Transmission Electron Microscopy

A reaction mixture containing the CVA9 (1.6 × 10^10^ PFU/mL), polyphenols (EGCG, 650 μM; RES, 880 μM), polyphenol-functionalized nanoparticles EGCG-AuNPs (Au concentration 164 μg/mL, ligand concentration 5.61 × 10^−1^ μM) and RES-AuNPs (Au concentration 530 μg/mL, ligand concentration 3.44 μM) in buffer (PBS-MgCl_2_) was incubated for 1 h at 37 °C and 5% CO_2_. During the incubation, the Formvar-coated copper grids were glow-discharged using the EMS/SC7620 Mini sputter coater (Electron Microscopy Sciences, Hatfield, PA, USA). Post-incubation, 5 μL of the reaction mixture was pipetted on the glow-discharged grids for 20 s and then blotted away using Whatman paper. To negatively stain the sample, 5 μL of 1% (*w*/*v*) Phosphotungstic acid (PTA in water, pH adjusted to neutrality) (Sigma-Aldrich, Shinagawa-ku, Japan) or 2% (*w*/*v*) Uranyl acetate (Electron Microscopy Sciences, Hatfield, PA, USA) was pipetted on grids and incubated for 30 s, and blotted away using Whatman paper. The samples were visualized in a JEOL JEM-1400 transmission electron microscope (JEOl, Tokyo, Japan). The diameter of functionalized gold nanoparticles from the images taken using the TEM was measured by automated counting using the BioImageXD software (www.bioimagexd.net, accessed on 16 January 2021).

### 2.14. Radioactive Binding Assay

A549 cells were cultured for 24 h in 96-well flat-bottomed microtiter plates at a density of 12,000 cells/well in 100 μL of DMEM supplemented with 10% FBS, 1% GlutaMAX and 1% penicillin/streptomycin antibiotics. On day 2, ^35^SMet/Cys-CVA9 (28,000 CPM corresponding to 1.38 × 10^8^ PFU/mL) was pre-treated with the polyphenols and polyphenol-functionalized nanoparticles EGCG (65 μM), RES (880 μM), EGCG-AuNPs (Au concentration 16.4 μg/mL, ligand concentration 5.61 × 10^−2^ μM) and RES-AuNPs (Au concentration 53 μg/mL, ligand concentration 3.44 × 10^−1^ μM) in buffer (PBS-MgCl_2_), and incubated at 37 °C for 1 h. Virus control without the compound was also prepared and incubated. Post-incubation, the sample tubes were placed on ice and 1% DMEM was added to make the final volume reach 100 μL. Following this, the samples with DMEM were added to the cells. Then, the 96-well plate was kept on ice while rocking for 1 h to ensure uniform binding of the virus on the cell surface. Next, the media was aspirated from each well and three washes with ice-cold PBS were performed to remove any unbound virus from the cell surface. To detach the cells, 150 μL of 1% Triton was added to the wells and incubated for 30 min. The entire liquid was transferred to a tube containing 4 mL of scintillation cocktail UltimaGold^TM^ and measured using the Tri-Carb^®^ 2910TR liquid scintillation analyzer.

### 2.15. Dilution and Temperature Assays

Dilution and temperature assays were performed using the CPE inhibition assay. For the dilution assay, CVB3 (4.4 × 10^8^ PFU/mL), polyphenol-functionalized nanoparticles EGCG-AuNPs (Au concentration 164 μg/mL, ligand concentration 5.61 × 10^−1^ μM) and RES-AuNPs (Au concentration 530 μg/mL, ligand concentration 3.44 μM) in buffer (PBS-MgCl_2_) were incubated at 37 °C for 1 h. An experimental virus control (without the compounds) and fresh virus control (without the compounds and without any incubation) were used as two controls for the assay. Post-incubation, each of the test samples and the experimental virus control were divided equally into two tubes, one was diluted 50 times using a buffer (PBS-MgCl_2_) and the other part was stored at −20 °C (as an undiluted sample). The diluted samples were incubated over 72 h at 37 °C, such that equal fractions from them were collected at regular time-intervals (1, 6, 24, 48 and 72 h) and stored at −20 °C. At the end of 72 h, all the diluted and undiluted samples were further diluted with DMEM and added onto the cells, to obtain a final MOI of 15. On the subsequent day, the plates were stained for CPE as explained before.

When testing the effect of temperatures, two approaches were used. One with longer and the other with shorter times of incubation. For testing the longer periods of incubation, CVB3 (4.4 × 10^8^ PFU/mL) was pre-treated with EGCG (650 μM), RES (880 μM), EGCG-AuNPs (Au concentration 164 μg/mL, ligand concentration 5.61 × 10^−1^ μM) and RES-AuNPs (Au concentration 530 μg/mL, ligand concentration 3.44 μM) in buffer (PBS-MgCl_2_) at RT (21 °C), 8 and 37 °C for 72 h. An experimental virus control (without the compounds) and fresh virus control (without the compounds and without any incubation) were used as two controls for the assay. Test samples and the experimental virus control were collected after incubating at the above-mentioned temperatures for 6, 24, 48 and 72 h and stored at −20 °C. After 72 h, all the samples were further diluted with DMEM and added to the cells, to obtain a final MOI of 15. On the subsequent day, the plates were stained for CPE, as explained before. In case of shorter incubation times, CVA9 (2 × 10^7^ PFU/mL) was pre-treated with EGCG (6.5 μM) at 21 or 37 °C for 1 min, 5 min and 1 h and then added to the cells.

### 2.16. Blind Docking of Polyphenolic Compounds against CVA9 and CVB3

#### 2.16.1. The Experimental Design

For the in silico docking experiment, the polyphenols Epigallocatechin gallate (EGCG) and Resveratrol (RES) were to be blindly docked over large external surfaces of the viral capsids of CVA9 and CVB3. Due to the overly large surfaces to be scanned, the capsids were segmented into relatively large grid boxes centered around their fold axes (2, 3 and 5) of rotational symmetry. This strategy was used in order to obtain overlapping segments spanning as much of the viral capsid surfaces as possible.

#### 2.16.2. Ligand Preparation

The molecular structures of the polyphenolic compounds, Epigallocatechin gallate (Drug Bank ID: DB03823) and Resveratrol (Drug Bank ID: DB02709), were reconstituted and protonated from their SMILE strings using the RDKit library. The MGLTools software [23] was then used to assign Gasteiger partial charges and merge non-polar hydrogen atoms.

#### 2.16.3. Capsid Preparation

Full capsids for the CVA9 Griggs strain (PDB ID: 1D4M) [24] and CVB3 Nancy strain (PDB ID: 6GZV) [25] were reconstituted from their first biological assemblies using the PyMOL software. While the CVA9 capsid was already protonated, the PDB2PQR tool was used to protonate the CVB3 capsid to pH 7. However, as this altered the chain labels and segment identifiers, an in-house Python script (capsidockprep.py) was designed to restore these labels. In order to screen the maximum outer surface area from the capsid, the grid box center was placed at each individual capsid fold axis of rotational symmetry, with the box y-z plane being perpendicular to the axis of rotation and partially embedded within the capsid. In order to do so, centroids (x, y and z coordinates measured in angstroms) for each of the fold axes were selected, before shifting the box along the *x*-axis by 10 Å to the right. Internally, before shifting, the capsid centroid is centered at the origin (0, 0, 0) prior to rotating the capsid centroid position vector to be along the *x*-axis. For simplicity, the PDB2PQR tool is wrapped within an in-house Python script to enable protonation when the “ph” parameter is set. Similar to AutoDock Vina, a grid interval size of 0.375 Å is used. The “gridsize” parameter is specified in grid points. The grid parameters for all of the fold axes are provided in Table 1. The AutoDock Vina plugin from PyMOL was used to visualize the effects of shifting and grid sizing.

The preprocessing by the in-house Python script greatly reduced the number of atoms by including protein chains belonging to any atom within the grid box, whilst keeping chain labels and segment IDs. Due to the high number of atoms required to capture the complete protomers in their protonated form for the 5-fold axis (beyond the limit of the PDB format), the experiment was focused around the 2- and 3-fold axes of rotation symmetry. As performed for the ligands, charges were assigned, and non-polar hydrogen atoms were merged using MGLTools.

#### 2.16.4. Capsid Ligand Docking

QuickVina-W [26] was used for docking the polyphenolic compounds, with a grid center of (0, 0, 0) and grid dimensions as defined in Table 1. The runs were performed at an exhaustiveness of 1000 with 24 cores per job, using GNU Parallel [27] at the Centre for High-Performance Computing (CHPC). A maximum of 20 top ligand binding poses were retained in each case run.

#### 2.16.5. Analysis of Docking Results

The Arpeggio tool [28] was then used to estimate the protein–ligand interactions around a subset of the complex trimmed at a maximum radius of 14 Å around the ligand atoms. An in-house Python script (without any external library dependency) was used to extract this subset, retaining the chain label and segment ID. Hydrogen bonding, hydrophobic interactions and the binding energy scores were then used to characterize ligand binding. The distributions of these measurements across the entire docking experiment were arranged according to the strains and the detected binding sites. In order to differentiate the binding sites from existing ones, both the canonical hydrophobic pocket [24] and the VP1-VP3 druggable interprotomer pocket (the Butcher Neyts’ pocket) from Abdelnabi and co-workers [25] were carefully mapped on all 3D structures, as their compositions and locations were not identical in each strain. In the case of the hydrophobic pocket, the PyVOL plugin [29] in PyMOL was used, specifying VP1 as the search space and a minimal pocket volume of 400 Å, in addition to the default parameters. The minimum volume had been determined beforehand using the palmitic acid-contacting residues (any protein atom within 4 Å of each ligand atom) from a 3D structure of the CVB3 Woodruff strain (PDB ID: 1COV) [30] to guide the cavity search. In the case of the interprotomer pocket, the PROMALS3D web server [31] was used to align the VP1 sequences and correct for any misaligned residue positions using 3D structural information. The ligand-interacting residues (binding the benzene sulfonamide derivative) from CVB3 (6GZV) identified by PDB were used as a reference in a sequence alignment with the CVA9 (1D4M) strain to obtain homologous positions for the latter [32]. No correction was needed for VP3 residues of the interprotomer surface, as the alignment contained no gaps. Unless specified otherwise, residues have been numbered according to the respective crystal structures.

### 2.17. Statistical Analysis

Statistical analysis was performed using GraphPad Prism 6 (GraphPad Software, San Diego, CA, USA). Data are presented as mean ± standard error (SEM). The 50% effective concentrations (EC_50_) and 50% cytotoxic concentrations (CC_50_) were calculated by regression analysis of the dose–response curves generated from the experimental data using the software. Statistical significance was calculated by performing one-way/two-way ANOVA followed by the Bonferroni test (* *p* < 0.05, ** *p* < 0.01, *** *p* < 0.001 and **** *p* < 0.0001).

## 3. Results

### 3.1. Epigallocatechin Gallate and Resveratrol Show Antiviral Activity on Enteroviruses

To evaluate whether selected polyphenolic compounds have antiviral effects on the chosen human enteroviruses (CVB1, CVB3, CVA9), we performed a cytopathic effect (CPE) inhibition assay on A549 cells. A high amount of CVB1 virus (10^7^ PFU/mL) was pre-treated with 10-fold serial dilutions of each compound for 1 h at 37 °C before adding to cells. Untreated virus and mock infection were used as positive and negative controls in the assay, respectively. The CPE analysis showed that Epigallocatechin gallate (EGCG) and Resveratrol (RES) as free molecules have a potent inhibitory effect on virus infection (Figure 1A,B). At a concentration of 6.5 μM of EGCG (Figure 1A, left) and 880 μM of RES (Figure 1B, left), CVB1 was efficiently inhibited. A significant toxicity could not be detected for any of the tested concentrations of the compounds (Figure 1A,B, right).

Next, we tested EGCG and RES bound to gold nanoparticles (Figure 1A,B, middle). The functionalized gold with EGCG or RES both showed antiviral activity. Nevertheless, RES-AuNPs were more efficient in inhibiting CVB1 infection than EGCG-AuNPs (RES ligand concentration 3.44 × 10^−1^ μM versus EGCG ligand concentration 5.61 × 10^−1^ μM). However, the necessary concentrations of ligand bound to AuNPs needed for virus inactivation were several orders of magnitude lower than the free polyphenols.

TEM images showed that the RES particles were smaller than the EGCG-AuNPs (Figure 1C, left). Automated determination of average core diameter from TEM images using BioImageXD software resulted in 131.5 ± 7.1 and 78.6 ± 2.3 nm for EGCG-AuNP and RES-AuNP, respectively. DLS analysis of the nanoparticles showed similar diameters as the TEM images, indicating that only one layer of polyphenol was attached. The zeta potential of the nanoparticles was −35 and −40 mV for the RES-AuNPs and the EGCG-AuNPs, respectively. The determination of gold content by ICP-MS showed a 3-fold higher concentration of gold for RES-AuNPs as compared to EGCG-AuNPs (Figure 1C).

We calculated the number of polyphenol molecules per AuNPs considering the nanoparticle diameter. Moreover, we assumed from the DLS versus TEM data that we have only one monolayer of molecules, which are oriented parallel to the surface, as observed previously for aromatic ring systems [17], and approximating the area of the polyphenol molecules from their molecule length and width, as measured by the Avogadro software. The calculations resulted in 47,239 EGCG molecules per AuNP and 19,122 RES molecules per AuNP. By considering the concentration of gold determined by ICP-MS and the diameter of the AuNP from TEM, the concentration of ligands was calculated to be 5.61 μM for EGCG and 34.4 μM for RES (see detailed calculations in the Appendix A). The calculated ligand concentrations were confirmed by an indirect quantitative analysis. For this analysis, EGCG and RES ligands present on the AuNP surface were replaced by an exchange with cysteine. The thiol group of cysteine has a stronger, quasi-covalent binding affinity to gold than the polyphenols. After incubation of EGCG-AuNP or RES-AuNP with a solution of cysteine, the remaining free thiol groups were determined photometrically using a colorimetric reaction with Ellman reagent. A 1:1 ratio of cysteine to polyphenol was assumed. The concentration of EGCG and RES determined by the Ellman test was 5.5 ± 0.1 μM for EGCG and 42.9 ± 1.1 μM for RES, respectively. In the following assays, we have used the calculated concentration.

#### 3.1.1. Detailed Antiviral Efficacy Studies

Once the preliminary screening results confirmed the antiviral activity of the polyphenols and the polyphenol-functionalized nanoparticles for CVB1, the antiviral efficacy on different enteroviruses was calculated as 50% effective concentrations (EC_50_) by regression analysis from their dose–response curves. The antiviral studies were performed using the CPE inhibition assay, where we incubated the compounds with high amounts of CVB3 (2 × 10^7^ PFU/mL) for 1 h at 37 °C. As shown in Figure 2A and Table 2, both polyphenols and the corresponding nanoparticles protected the A549 cells from CVB3 infection in a dose-dependent manner. The EC_50_ value for EGCG was 3.411 μM, 32 times more effective than RES (107.894 μM). Interestingly, the ligands bound to AuNPs (0.051 μM for EGCG and 0.004 μM for RES) were 67 or 26,974 times more potent than the free polyphenols, respectively. The RES-AuNPs were in this case 13 times more potent than EGCG-AuNPs against CVB3 infection.

We also tested the compounds for inhibition of other enteroviruses, using highly concentrated CVB1 (2 × 10^7^ PFU/mL) and CVA9 (2 × 10^8^ PFU/mL) enteroviruses. Figure 2B,C and Table 2 show that EGCG is 3 and 13 times more effective for CVB1 and CVA9 respectively, than for CVB3. In general, the RES ligand was less effective compared to the EGCG ligand. The results also show that the ligands have better efficacy when functionalized on the gold nanoparticle for RES. Depending on the virus and the ligand, there was a 5- to 26,974-fold improvement in efficacy (Table 2). Usually, the improvement in efficacy was better for RES than for EGCG. This can be due to an increased solubility of RES when bound on nanoparticles.

Since the polyphenols and functionalized AuNPs both exhibited potent antiviral activity, we tested if the inhibitory effect was due to bound ligands and not due to free ligands present in the solution. This was confirmed by pelleting the functionalized AuNPs by ultracentrifugation and testing the pellet for the antiviral activity. The antiviral activity of the NP solution was similar to the activity in the resuspended pellet of AuNPs, confirming that antiviral efficacy was not related to detached ligands (Appendix A).

Detailed cytotoxicity studies using the CPE inhibition assay on uninfected cells were repeated under the same conditions as used for the antiviral studies (Figure 2D). Mock infection, having no virus or compounds, was used as a control for the assay. As shown in Figure 2D, none of the compounds reduced the viability of A549 cells at the effective antiviral concentrations, thus verifying the antiviral potential of the compounds. The 50% cytotoxic concentrations (CC_50_) calculated by regression analysis from the dose–response curves are shown in Table 2. The results showed no toxicity for RES or for both AuNPs, while for EGCG, we detected toxicity at a very high concentration (CC_50_: 2.218 mM). Thus, the selectivity index (SI) deduced from the ratio of CC_50_/EC_50_ is incalculable for compounds other than EGCG. For EGCG, the SI values are very high for all tested viruses (Table 2). Based on the high (or incalculable) SI values, it can be said that these compounds effectively protect the A549 cells from the tested enterovirus infections with no toxicity.

#### 3.1.2. EGCG and RES Show Long-Term Antiviral Efficacy at Different Temperatures

To investigate the antiviral efficacy of the polyphenols further, we performed antiviral tests for the functionalized AuNPs at different temperatures and different periods of incubation (Figure 3A). First, the functionalized AuNPs were pre-incubated with the virus at 8, 21 and 37 °C. Untreated viruses in PBS-MgCl_2_ buffer incubated at the same temperatures were used as an experimental control. Virus infectivity was measured after incubating the polyphenols with viruses for 6, 24, 48 and 72 h using the CPE assay. The results showed that the observed effective concentrations of EGCG-AuNPs (ligand concentration 5.61 × 10^−1^ μM) and RES-AuNPs (ligand concentration 3.44 μM) inhibited infection at the tested temperatures, at least for the whole period for which the untreated virus was showing infectivity. Namely, in the control experiment with the untreated virus, it was observed that the virus infectivity was decreasing with time, more quickly at higher temperatures, as expected.

The experiments were repeated with the ligands at 21 and 37 °C (Figure 3B). The results were similar to those with AuNPs. Interestingly, RES could not rescue cells from infection before 6 h at 21 °C, but was already quite effective after 24 h. We then tested whether the antiviral efficacy was lost if the solution was diluted after the initial incubation with the virus (Figure 3C). The hypothesis was that strong dilution may disconnect the virus and ligand or ligand-NP and lead to a re-established infectivity. After 1 h of incubation of the virus with functionalized AuNPs at 37 and 21 °C, we diluted the solution 50 times and continued the incubation. Samples were collected from diluted solutions at different time points (1, 6, 24, 48 and 72 h) and tested for infectivity on A549 cells. The CPE analysis of diluted and control virus samples (normalized to contain comparable amounts of virus) showed that the 50-times diluted samples maintained the antiviral activity very well, and similarly to undiluted samples (Figure 3C). Similar results were observed with EGCG and RES without AuNPs even when diluted (data not shown).

We also wanted to understand the effectiveness of polyphenols at shorter incubation time intervals at 21 and 37 °C (Figure 3D). The results showed that already, 1 min of incubation slightly increased the cell viability, as indicated by reduced lytic viral activity. This effect was more apparent at 37 °C. After 5 min of incubation, cell viability increased strongly at both temperatures, and 1 h of incubation already showed full protection. These results thus confirm that the polyphenols show good antiviral efficacy at various temperatures and after a short contact time.

### 3.2. Mechanism of Action

To understand if the compounds acted as inhibitors for virus attachment or in other steps of infection, we performed a simple time of addition study. In the assay, cells were first infected with CVB3 for 1 h at 37 °C and then the compounds were added, followed by overnight incubation. As shown in Figure 4A, polyphenols did not exhibit any antiviral activity when added 1 h post-infection (p.i.). This excludes antiviral action on intracellular processes.

Three possible mechanisms of pre-entry action were studied: (1) the polyphenols and polyphenol-functionalized nanoparticles stabilize the virus and prevent its opening (and release of genome), (2) the compounds prevent binding of the virions to the cell surface and (3) the capsid is disrupted or damaged by the polyphenols, and thus, the genome is prematurely released from the virions.

A potential increase in capsid stability of the virions was tested in a thermal stability assay, PaSTRy [15,22]. In the assay, gradual heating of the virus up to 90 °C induces the uncoating and release of the viral RNA to the solution, typically between 40 and 50 °C for enteroviruses [22]. Non-fluorescent SGII in the medium intercalates within the double-stranded regions in the viral RNA and emits fluorescence upon binding [33]. Increase in capsid stability would lead to a higher melting temperature, whereas lower melting temperature would suggest destabilization.

Untreated CVB3 in a storage buffer (PBS MgCl_2_) had a transition temperature of 42 °C (Figure 4B), while in an opening buffer, the fluorescence was maximal already without heating [16,21]. The opening buffer promotes virus opening by releasing the pocket factor with a combination of albumin and low sodium/high potassium content. When 0.5 µg of CVB3 was pre-incubated with a concentration of 65 µM EGCG, strikingly, low fluorescence was recorded and no transition could be observed in the tested temperature range, probably because EGCG stabilized the virion and prevents genome release (Figure 4B). If the opening buffer was added to the EGCG-treated virus, the initial fluorescence was higher but still lower than in the absence of EGCG and, again, no significant transition was observed during the heating.

To study the mechanism of this unexpected stabilization in detail, we used our previously developed real-time spectroscopy uncoating assay, which can detect the viral genome at various stages of uncoating in vitro at physiological temperature, 37 °C. This assay relies on the fact that SGII is able to bind to the virus RNA inside the expanded capsid and outside the broken capsid, but the dye is excluded from the intact capsid [16,21]. Thus, this approach enables us to detect the expanded, primed state and the fully opened state of the virus, while the intact virus would show no fluorescence. For this assay, CVB3, polyphenol-functionalized nanoparticles and SGII were mixed in a buffer, and the fluorescence was measured. The control virus in the storage buffer revealed a low background fluorescence that did not change much during the 1 h experiment at 37 °C, suggesting that the virus is relatively stable at 37 °C in this buffer (Figure 4C), as expected from earlier studies [16]. The assay includes a subsequent addition of RNAse to the samples that degrades the released RNA. The RNase molecule is too large (72 kDa) to enter the primed virus particles and digest the viral genome, hence allowing to distinguish fully open capsids from partially damaged ones. SGII is small enough (0.45 kDa) to enter, bind to the genome and emit fluorescence [15]. The values after RNAse addition were subtracted from the fluorescence values without RNAse, thus providing the net RNA released from the viruses in the assay. The presence of EGCG-AuNPs or RES-AuNPs caused an even lower overall fluorescence compared to the control virus sample in this assay, suggesting that the functionalized AuNPs prevented expansion that normally occurs to some extent in the storage buffer at 37 °C. The fluorescence did not change with time, confirming that the functionalized AuNPs prevented opening and RNA release. The results were similar if the free polyphenols were added instead of polyphenol-functionalized AuNPs (data not shown).

To test whether the presence of ligands also prevents binding of the virus to the surface receptors of A549 cells, we performed a binding assay using a metabolically radiolabeled virus. Radioactive ^35^SMet/Cys-CVA9 was first incubated with the compounds for 1 h, followed by binding them to A549 cells for another 1 h on ice, and continuing by thorough washings. Measurement of the radioactivity from the virus bound to the cell surface in the presence of the compounds revealed that both the ligands and the functionalized nanoparticles drastically reduced binding of the virus to the cell surface (Figure 4D). The results altogether suggest that the EGCG, RES and their functionalized AuNPs interact directly with the capsid, stabilize it strongly and hence prevent the infection by both preventing uncoating and receptor binding.

#### Polyphenols Cause Clustering and Aggregation of the Enteroviruses

With a sucrose gradient, viruses can be separated into intact, empty and expanded/intermediate particles [15,16]. Such a separation was performed with the radioactive ^35^SMet/Cys-CVA9, pre-incubated with EGCG or RES for 1 h at 37 °C (Figure 5A). Fresh virus control without the polyphenols and without preliminary incubation at 37 °C showed a typical single peak of intact viruses covering about 4–5 fractions in the bottom part of the gradient. Incubation of the control virus in the storage buffer for 1 h at 37 °C showed no significant difference to the fresh virus. Only a small peak broadening can be observed due to the presence of some expanded virions [15]. Strikingly, when the virus was treated with the polyphenols, there was a complete loss of radioactivity. We assume that the clustering of the virus induced by the polyphenols led to aggregates, which precipitated or adhered to the walls of the tubes. Myllynen et al. [15] observed a similar loss of radioactivity when vulnerable virus preparations were aggregated during fractionation in a CsCl_2_ gradient.

To test our hypothesis, we pretreated the virus with polyphenols or polyphenol-functionalized AuNPs for 1 h and then visualized them under the transmission electron microscope. Figure 5B shows negatively stained control viruses and viruses treated with polyphenols and polyphenol-functionalized AuNPs. Control viruses mostly showed single viruses distributed homogenously over the grid. Intact virus particles have the stain around the capsid and have a bright center, whereas those that are expanded, or broken/empty allow the stain to enter the capsid and have a darker appearance or dark center, respectively (some empty virions are indicated in Figure 5B). In the presence of the polyphenols or polyphenol-functionalized AuNPs, virus particles appeared only in large aggregates. Interestingly, the viruses inside the aggregates seemed to be intact, and empty virus particles were rarely seen. However, some uneven coloring of the virions was observed. These results confirmed that, in addition to stabilization of the particles, the polyphenols cause clustering of the viruses.

To observe the clustering of the viruses in a more robust way in solution, we performed DLS after EGCG treatment. We detected the virus in DLS with a hydrodynamic diameter of ~22 nm, which is a little lower than expected for CVA9 (Figure 5C). However, this peak could be detected in a very reproducible manner, and we observed only a few aggregates in the virus samples by DLS. After treating the virus samples with EGCG, the virus-associated peak completely disappeared, and particles were detected only in the size range of 650–1240 nm, further confirming that, indeed, polyphenols cause aggregation of the enteroviruses. The virus was then diluted using 10-fold dilutions while keeping the EGCG concentration constant. The aggregates appeared upon every dilution, indicating that a high concentration of viruses was not a prerequisite for aggregate forming. A similar size distribution was observed in all tested dilutions.

We also studied if the clustering with a lower titer of the virus would still cause a similar drop of infectivity. We did this by drastically reducing the CVB3 concentration to 2 × 10^5^ PFU/mL, which is 100 times less than in the previous experiment (Appendix A). Despite the dilution of 100 times, we observed an efficient antiviral effect.

### 3.3. Discovery of Multiple Ligand Binding Sites on the Surface of CVA9 and CVB3

The docking poses were visualized from the 2-fold (Figure 6) and 3-fold (Figure 7) axes of symmetry, respectively. From the PyVOL calculations, the hydrophobic pockets (HPs) were visibly smaller in CVB3 (subfigures B and D from Figure 1 and Figure 2) compared to CVA9 (subfigures A and C from Figure 6 and Figure 7). More specifically, in CVB3, the buried HP site was found to be composed of VP1 residue positions 93, 95, 190–194, 205–212 and 258–261 (residue numbering according to 6GZV). The analogous pocket in CVA9, however, consisted of residues 95–98, 115, 117, 144, 146, 170, 181, 183, 186–188, 192, 193, 210, 212–216 and 219 in CVA9 (residue numbering according to 1D4M). Abdelnabi and co-workers’ work describes the Butcher-Neyts pocket that comprise VP1 residue positions 73, 76, 77, 78, 234 and 236, and VP3 residue positions 234–236 in CVB3 [25]. While CVA9 and CVB3 shared identical residue loci for the VP3 component of the Butcher-Neyts pocket, CVB3 had VP1 mutations M76F and E77T and VP3 mutations D234Q and K236F (using CVA9 as a reference). EGCG was the only compound found to bind the Butcher-Neyts pocket in CVA9, albeit with dissimilar contact residues, consisting of D158, D159, Y160, W162, Q163 and R222 from VP1, and the VP3 residues Q233, D234, N235 and R237 (Figure 7A). In our experimental set-up, the Butcher-Neyts pocket was more completely exposed within the docking grid box covering the 3-fold axis protomers, which explains the absence of bound ligands within the grids designed for the 2-fold axis of symmetry.

Four novel binding sites were found, each with the support of more than one ligand pose, and were named S1, S2, S3 and the 3-fold axis pore, based on consensus residues shared across ligand-contacting surfaces. A fifth one—tentatively named S4—had a lower level of support, as it was bound only once. S1 was detected only in CVA9 and comprised the VP1 residues Y208, G209, N211, F262, S263, V264 and D265, and VP2 residues M132, G133, G134, G138, Q139, A140 and F141, from a consensus of all 16 detected ligand poses at that site. S2 was present in both CVA9 and CVB3. In CVA9, S2 comprised VP1 residues F202 and T283, and VP3 residues R86, L87, Q88, P137, G138, A139, Q181, D182, E183, T185, S186, A187, G188 and Y189, and was derived from two poses. In CVB3, S2 is composed of VP1 residue F199 and VP3 residues P86, G140, A186, G187, G188 and F189, obtained from a consensus of residue contacts derived from 12 ligand poses. The 3-fold axis pore was detected in the CVA9 capsid only in the case of RES (Figure 8). Its consensus ligand-contacting surfaces (derived from four poses) are composed of VP2 residues R12, E27, K116, R193 and T194, VP3 residues A124, A126, T127, I158, G159 and L160 and the VP4 residue L68 in CVA9. In the same figure, the three ligand binding sites subtending the 3-fold axis with a similar rotational symmetry provide support for an actual binding site. Upon closer inspection, these sites were found to be connected, as part of a larger cavity. Access to this hollow space in CVA9 is lined with residues such as methionine (VP3), phenylalanine (VP2) and aspartic acid (VP3). The fact that these all have titratable side chains suggests that pH may play a role in driving the pore dynamics.

For a closer inspection of the non-bonded interactions occurring for RESV and EGCG, single ligand poses docked at each of the newly found sites are depicted in Figure 8. Their favorable binding energy scores and multiplicity of binding locations, combined to the formation of multiple H-bonding interactions, provide good support for the stability of these compounds for each of the Coxsackievirus strains. For instance, using the binding locations described in Figure 6 and Figure 7, EGCG forms four H-bonding interactions involving VP1 and VP2 in CAV9 at S1, two H-bonds at S2 in CBV3 involving VP1 and VP3, three H-bonds at S3 in CBV3 involving VP1 and VP3 and five potential H-bonds at S4 in CAV9. Similarly, RES forms five potential H-bonding interactions with residues from VP2 and VP3 at the 3-fold axis pore in CAV9. While there are many more ligand binding poses from both the strains and binding sites, only one pose is shown per site, as an example.

#### RES and EGCG Show Differential Binding Patterns for CVA9 and CVB3

RES has fewer H-bond acceptors compared to EGCG, in addition to having a smaller size and larger hydrophobicity, as detailed in the Discussion Section. All docked poses (across ligands and strains) are summarized in Figure 9 in terms of ligand binding energies, their aggregated number of H-bonds and hydrophobic interactions at each of the six sites (S1, S2, S3, 3-fold, Butcher-Neyts pocket and HP). Our results suggest that there are both specific and shared binding sites for EGCG and RES, with strain also playing a role. Ligand binding at the sites S1 and the pore found at the 3-fold axis was only noted in CVA9. The S1 site was bound by EGCG only with CVA9. In the case of S2, EGCG showed strain bivalence, although the calculated affinity was slightly higher in CVA9. RES also bound to S2, but only in the CVB3 strain. RES displayed slightly higher estimated affinity compared to EGCG for the same site. Binding to the S3 site was achieved only by EGCG in CVB3. Binding at the pore found at the 3-fold axis was only observed in the case of RES, only in CVA9. EGCG was found to be bound to the Butcher-Neyts pocket only in CVA9. HP was bound by both compounds in both strains. The most favorable binding energies were recorded for both EGCG and RES in the HP of CVA9, most likely due to the shape and partial charges present within the deeper hydrophobic tunnel. RES however displayed the highest affinities for the HP, which suggests more favorable binding poses due to the linear ligand shape. Both compounds performed worse in the CVB3 HP compared to CVA9, most likely due to the shallower hydrophobic pocket. The binding energies recorded at sites S1, S2, S3 and 3-fold axes are generally lower than those observed in the HP, most likely due to a decreased exposure to protein surfaces.

Overall, EGCG bound to more sites in both CVA9 and CVB3. However, RES showed high selectivity for the hydrophobic pocket. The high number of binding sites for both compounds (compounded by the capsid symmetries) may entail a noticeable dysfunction of viral capsid behavior, especially more so in CVA9 due to the comparatively higher number of sites and the lower binding energies for the HP region. This also explains the difference in inhibition efficacy of EGCG as compared to RES.

## 4. Discussion

There is a great need for broad-acting non-toxic antivirals that reduce the virus load in the body or on surfaces. Present and future pandemics as well as the yearly occurring epidemics cause high economic costs due to hospitalizations and absence from workplaces and from schools. Enteroviruses still lack a clinically accepted antiviral for treatment or disinfection of hands and surfaces. Therefore, we studied two polyphenols, EGCG from green tea and RES from red grapes and other fruits, as soluble polyphenols and polyphenol-functionalized gold nanoparticles and explored the mechanism of action for the molecules on three very stable, non-enveloped enteroviruses (CVA9, CVB1 and CVB3).

Our results showed that EGCG and RES have strong antiviral efficacy against the three enterovirus serotypes. Antiviral efficacy in general was already known for the two polyphenols before [34], but not for enteroviruses. The EC_50_ values were in the range of 0.266–3.411 µM for EGCG and 107.894–330.369 µM for RES, thus suggesting a good efficacy already at low concentrations. The CC_50_ showed no cytotoxicity. The polyphenols bound to the AuNPs showed EC_50_ values in the range of 0.004–0.237 µM for EGCG and 0.004–0.068 µM for RES.

Most antivirals target cellular replication steps of the viruses, such as promoting apoptosis and via inhibiting Nf-kB or MAPK pathways and replication [5,35,36]. Most studies on the polyphenols as antivirals for enteroviruses focus on EV71 and observe antiviral action past entry and during replication of the viruses. Our results reveal that both EGCG and Resveratrol act as entry and attachment inhibitors by binding to enterovirus surfaces. This was confirmed by the time of addition CPE assay when polyphenols pre-incubated with viruses showed inhibition, while drugs added after entry to cells had no effect. In silico docking data further showed that these polyphenols could bind to multiple areas on the virion surface. Other drugs binding directly to enterovirus capsids, such as pleconaril, target the hydrophobic pocket that is normally housed by an aliphatic lipid such as palmitic or oleic acid [37,38]. The hydrophobic pocket has been associated with capsid stability and suggested to release its lipid upon viral RNA release [39]. We have recently shown that albumin treatment of enteroviruses Echovirus 1 and CVA9 do expand the virion, which is typically an event associated with emptying of the pocket. However, we showed that treatment with albumin, which may steal fatty acids from the virion, does not yet lead to virus uncoating but rather virus priming to an intermediary form, leading to virus RNA release only upon further cues, such as changes in concentration of key ions [16,21]. Here, we have studied three viruses, two of which show similarly spacious pockets, whereas the CVB3 Nancy strain has a collapsed pocket that cannot house large molecules. Consequently, CVA9 showed the highest estimated binding energies for both drugs in the HP. In addition, simulations showed that the HP is the preferable docking site for the drugs with the 73% of the docking poses associated with it.

A novel algorithm that was developed for this study to reveal the docking of the molecules on enterovirus surfaces considered larger areas around the 2- and 3-fold symmetry axis. Remarkably, the docking poses for the polyphenols, RES and EGCG on the viral capsids suggested the existence of multiple binding sites, as seen from the 2-fold (Figure 6) and 3-fold (Figure 7) axes of symmetry, respectively. It seemed that the binding locations were influenced by both the physico-chemical characteristics of the polyphenols, the geometric landscapes and partial charge distributions present within the capsid assemblies for each strain. While both compounds are polyphenolic, Resveratrol is classified by PubChem as smaller and more hydrophobic (2 heterocycles, an XLogP3 value of 3.1 and 3 H-bond acceptors) than EGCG (4 heterocycles, an XLogP3 value of 1.2 and 11 H-bond acceptors) [40], which we believe to be a main factor influencing the distribution of the ligand binding sites. The fact that the HPs detected in CVA9 are much deeper than those of CVB3 may explain the better fit of the smaller Resveratrol further inside the capsid compared to EGCG. Altogether, three new sites were found, in addition to the known HP and the recently discovered pocket that we call the Butcher-Neyts pocket [25].

Our results suggest that there are both specific and shared binding sites for EGCG and RES, with the strain also playing a role. HP showed good binding by both viruses and both drugs, suggesting that it is the main attraction on the virion surface. Instead, S1 was only detected with EGCG in CVA9, while EGCG bound S2 in both CVA9 and CVB3. Resveratrol bound to S2 only in CVB3 but to the 3-fold axis only in CVA9, further suggesting that the structural differences of the drugs as well as of the virions dictate the docking to the virions.

The predicted binding energies were lower in the newly found docking sites in comparison to the canonical HP, probably due to its deep invagination and charges present in the pocket. In silico-determined binding energies only provide a general estimation of the affinity of ligand binding and are generally useful for doing preliminary ranking of docked ligands. However, the frequency of binding to a site can additionally assist in the evaluation of lower-scoring poses. Out of the 159 docking poses, the HP ranked first with 73% of poses assigned to it, followed by S1 (10%), S2 (9%), the Butcher-Neyts pocket (4%), the pore at the 3-fold axis (3%) and the S3 site (1%). As the Neyts pocket has already been independently proven experimentally, there are valid reasons to believe that S1, S2 and the 3-fold pore are actual novel binding sites, based on the closeness in percentage clustering, a partial overlap in the range of binding energies and the number of H-bonding interactions. There is less support for the S3 site and S4, even though both may occur.

It is no wonder that the multiple ligand binding sites on the capsid led to a strong antiviral effect directly on the virion. Our studies on the molecular mechanisms of the action showed that stabilization rather than premature RNA release occurred for enteroviruses. This was shown by several methods: (1) thermal assay, (2) our previously developed real-time uncoating assay, (3) radioactive gradient assay and (4) TEM. Stabilization and prevention of virus uncoating was evident even at higher temperatures. Our results also showed that ligand binding was accompanied by strong clustering of the virions in addition to prevention of virion uncoating in the clusters.

The fact that EGCG is a bigger molecule with more hydroxy groups can be the reason why, in comparison to RES, it is more efficient in inhibiting the enteroviruses. The number of hydroxy groups has been addressed before for the difference in antiviral action of the polyphenols, apigenin and luteolin [36]. Interestingly, Resveratrol being a nearly flat molecule (Appendix A) is more closely attached to the surface of a gold nanoparticle. Previous experiments indicate that aromatic rings are oriented parallel to the gold surface [17]. EGCG has a nearly planar aromatic ring system with one ring perpendicular to the base (Appendix A).

This indicates that partial immersion of the EGCG is possible and can be responsible for the increase in stability and the subsequent virus aggregation. On the other hand, the three-dimensional structure of the free EGCG molecule can cross-link several viruses by immersing into their binding pockets and inducing cross-linking of two viruses, explaining why the free EGCG is more efficient than the RES molecule. RES is six times more concentrated when bound to AuNPs as compared to EGCG. Moreover, the molecules are presented on smaller nanoparticles that have a strong surface curvature. Even if we assume that RES only penetrates slightly into the pockets because it is nearly parallel to the surface, the sheer number of interactions between virus and RES and the RES presentation by the more curved surface of nanoparticles can work as a cross-linker between multiple binding pockets on several viruses. This could explain why RES bound to nanoparticles is more efficient than the free molecule.

Another interesting fact that we observed in our infection experiments was that RES was slower to inhibit virus infection. After 6 h, its antiviral efficacy was still moderate, whereas after 24 h and later, it had reached full efficiency, while EGCG was very efficient already at earlier time points. This may be directly because RES has fewer docking sites on the virion surface. However, it may also be because it has less hydroxylic groups on its structure, leading to lower binding affinity.

Altogether, the two studied polyphenols showed great antiviral efficacy for all three studied enteroviruses. The discovered mechanism of action is that the polyphenols immerse in several binding sites on the virion surface, resulting in stabilization of the virions against opening, even in harsh conditions. In addition, formation of large clusters was observed, which might be another efficient way to lower the incidence of virions entering cells. Clustering and the resulting precipitation of the virus–polyphenol or polyphenol–AuNP clusters indicate a potential use for safe water clearing applications and for decontamination of surfaces. Future studies will focus on clarifying if the clustered viruses will be prevented from penetrating the mucus layer or be used as vaccines. Previous studies have also demonstrated the therapeutic potential of polyphenols such as EGCG and Curcumin, for treating protein aggregation diseases associated with peripheral neuropathy [41]. The mechanism by which EGCG works, to prevent the disease progression in case of amyloid fibril formation, is by stabilizing the protein after binding to it and forming soluble nontoxic aggregates [42,43]. EGCG interacts with amyloid-forming proteins through hydrophobic interactions or H-bonds, forming stable oligomers [44]. Similar interactions between the virus capsid and polyphenols have been described in our computational studies. Whether the mechanism behind virus stability/loss of infectivity and prevention of formation of amyloid aggregates reported previously is the same remains to be shown.

In general, it can be concluded that especially the AuNP-bound polyphenols might have interesting potential in future applications for (i) antiviral surface coatings, which can already reduce the viral load before it is transferred to the body surface and unintentionally up-taken, and (ii) as oral antivirals. The oral administration of the polyphenol-coated nanoparticles is especially interesting as the nanoparticle and the bound polyphenol are localized in the primary infection site for enteroviruses, in the gut, before they likely pass through the intestinal cell layer and hence can significantly reduce the viral load of the intestine. The systemic toxicity of these nanoparticles will be determined in future animal experiments. It requires consideration as the polyphenol-AuNPs due to their size and nearly neutral surface charge have optimal properties to penetrate the mucus layer of the intestinal tract [45,46].

## 5. Conclusions

Based on our findings, EGCG and RES effectively inhibit enterovirus infection, even after short encounters with the viruses and at various temperatures. Interestingly, the studied polyphenols have better efficacy when functionalized on gold nanoparticles, particularly for RES. The efficacy of the compounds was not compromised even when diluted 50 times. Mechanistic studies revealed that the polyphenols and polyphenol-functionalized AuNPs (i) cause clustering and stabilization of the virions, (ii) prevent the release of viral RNA and (iii) prevent the virus from binding to the cell surface. In silico docking experiments of these molecules against 2- and 3-fold symmetry axes of the capsid identified three novel binding sites in addition to the canonical hydrophobic pocket, and the Butcher-Neyts pocket. The presence of several binding sites for the polyphenols explains the strong increase in the stability. The strong clustering and inactivation of the virions may turn out to be helpful for future applications, for example, vaccination.

## Figures and Tables

**Figure 1 pharmaceutics-13-01182-f001:**
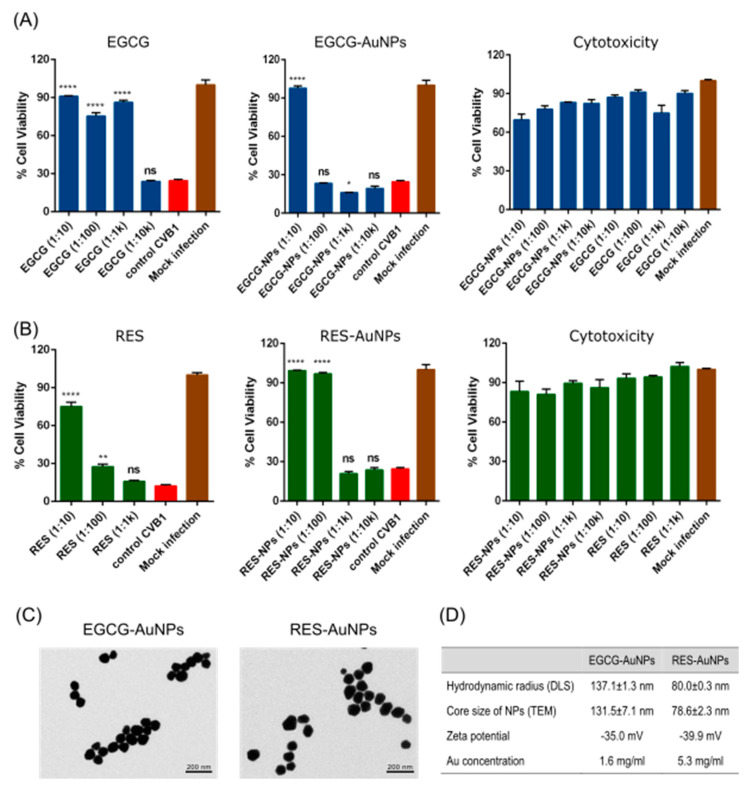
Polyphenols and polyphenol-functionalized nanoparticles block virus infection. (**A**) Epigallocatechin gallate (EGCG) and (**B**) Resveratrol (RES) and polyphenol-functionalized nanoparticles (AuNPs) were screened against CVB1 using the CPE inhibition assay to evaluate infection and cytotoxicity of the polyphenolic compounds. The test samples and the virus control are normalized against the mock infection. N = 3 replicates for each dilution of the samples tested. Average values + standard error of the mean (SEM) are shown. * *p* < 0.05, ** *p* < 0.01and **** *p* < 0.0001 versus the virus control (analyzed using the one-way ANOVA with Bonferroni tests). (**C**) Transmission electron microscopy images of EGCG-AuNPs and RES-AuNPs. (**D**) Physico-chemical characteristics of the polyphenol-functionalized AuNPs. The mean core size of the nanoparticles was calculated using the BioImage XD software by processing TEM images containing 684 (EGGC-AuNPs) and 2411 (RES-AuNPs) nanoparticles, respectively. The results of the TEM and DLS are expressed as average values ± standard error of the mean (SEM). Hydrodynamic size and zeta potential were evaluated from three separate readings using a Malvern Zetasizer. Au concentration was obtained using ICP-MS. Scale bars = 200 nm.

**Figure 2 pharmaceutics-13-01182-f002:**
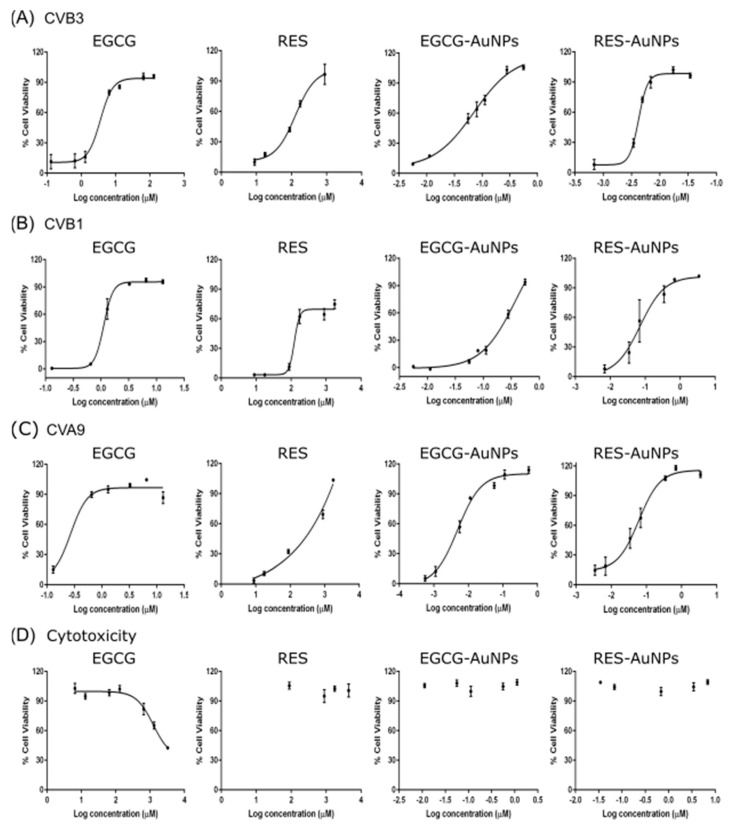
Dose–response curves for studying the antiviral efficacy of Epigallocatechin gallate (EGCG) and Resveratrol (RES) and polyphenol-functionalized nanoparticles (AuNPs), against (**A**) CVB3, (**B**) CVB1 and (**C**) CVA9 and their (**D**) cytotoxicity, using the CPE inhibition assay. Drug concentration of the compounds are shown as Log (10) of µM on the *x*-axis. The results are mean of two independent experiments and are expressed as average values ± standard error of the mean (SEM).

**Figure 3 pharmaceutics-13-01182-f003:**
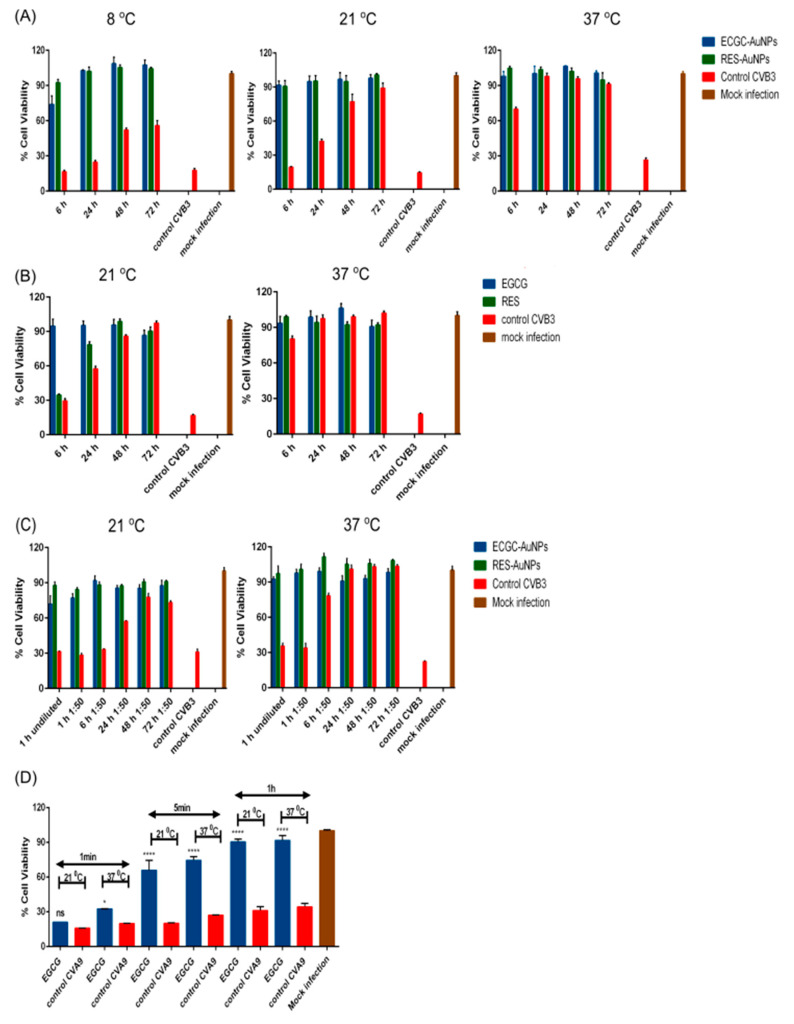
Effect of time and temperature (**A**,**B**,**D**) and dilution (**C**) on antiviral activity of polyphenols and polyphenol-functionalized nanoparticles (AuNPs). (**A**,**B**) EGCG-AuNPs (ligand concentration 5.61 × 10^−1^ μM), RES-AuNPs (ligand concentration 3.44 μM), EGCG (650 μM) and RES (880 μM) were incubated with CVB3 (4.4 × 10^8^ PFU/mL) at different temperatures over 72 h. Sample fractions were collected at 6, 24, 48 and 72 h and a CPE was performed from them. (**C**) The effect of dilution on the antiviral activity of polyphenol-functionalized nanoparticles (AuNPs) after an initial preincubation of 1 h between CVB3 (4.4 × 10^8^ PFU/mL) with EGCG-AuNPs (ligand concentration 5.61 × 10^−1^ μM) and RES-AuNPs (ligand concentration 3.44 μM). (**D**) EGCG (6.5 μM) and CVA9 (2 × 10^7^ PFU/mL) were incubated at different time intervals (1 min, 5 min and 1 h) and at different temperatures (21 and 37 °C). Results (**A**–**D**) are expressed as the number of viable cells left, detected by crystal violet staining. Every experiment had an experimental virus control that was used to assess the infectivity of the virus at each point and a mock infection showing the status of viable cells. Experiments 3A and 3B were performed twice with *n* = 3, and experiments 3C and 3D were performed once, with *n* = 3 and *n* = 2, respectively. The results are expressed as average values + standard error of the mean (SEM). Experiments 3A, 3B and 3C were analyzed by two-way ANOVA and 3D by one-way ANOVA, followed by the Bonferroni test (* *p* < 0.05, and **** *p* < 0.0001).

**Figure 4 pharmaceutics-13-01182-f004:**
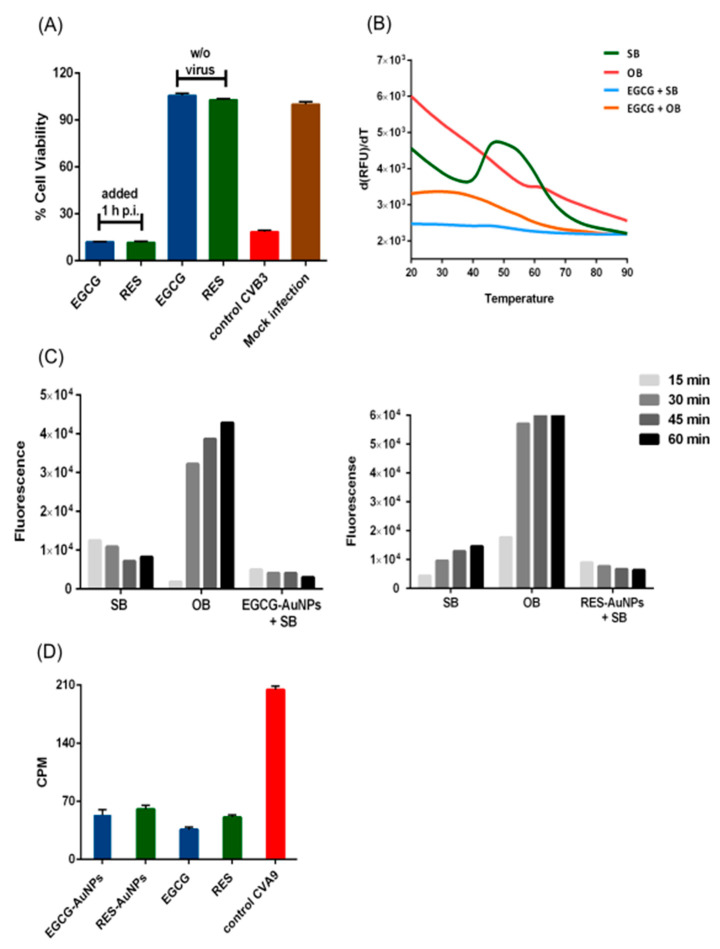
Studies on mechanism of antiviral activity of polyphenols and polyphenol-conjugated nanoparticles (AuNPs). (**A**) Time of addition assay, where the polyphenols EGCG (65 μM) and RES (880 μM) were added to cells 1 h post-infection. The results are the mean of two independent experiments, with *n* = 5. Average values + standard errors of the mean (SEM) are shown. (**B**) The PaSTRy assay measures the thermal stability of CVB3 (0.5 µg) in the presence of the EGCG (65 μM). The tests were performed in the presence of the storage buffer (SB; PBS with 2 mM MgCl_2_) and opening buffer (OB; 20 mM NaCl, 6 mM KH_2_PO_4_, 12 mM K_2_HPO_4_ and 0.01% faf-BSA), depicted as blue and orange color, respectively. This is a representative result from three experiments, with *n* = 3. (**C**) The Uncoating assay shows that EGCG-AuNPs and RES-AuNPs lower the release of the viral RNA from virions and prevent expansion of the virions. The assay is based on real-time fluorescence spectroscopy, which detects high fluorescence of SGII when it is bound to the viral genome, either free RNA released from the virions or virion-bound RNA when SGII can enter an expanded virus. Here, CVB3 (0.5 μg) was incubated with SB or OB as controls or with SB in the presence of EGCG-AuNPs (ligand concentration 5.61 × 10^−2^ μM; left graph) or RES-AuNPs (ligand concentration 3.44 × 10^−1^ μM; right graph). The fluorescence values have been subtracted with the measurement of RNAse in the assay, and the difference between the calculations, which is the net release of RNA, has been plotted in the graphs. Both the graphs are a representative result from three experiments. (**D**) The binding assay measures the binding of metabolically radio-labeled virus onto the cells in the presence of compounds on ice. Control CVA9 is the binding measured from the virus without polyphenols. Average values of radioactivity (CPM, counts per minute) + standard error of the mean (SEM) are shown from one representative experiment that was performed twice with similar results, with *n* = 3.

**Figure 5 pharmaceutics-13-01182-f005:**
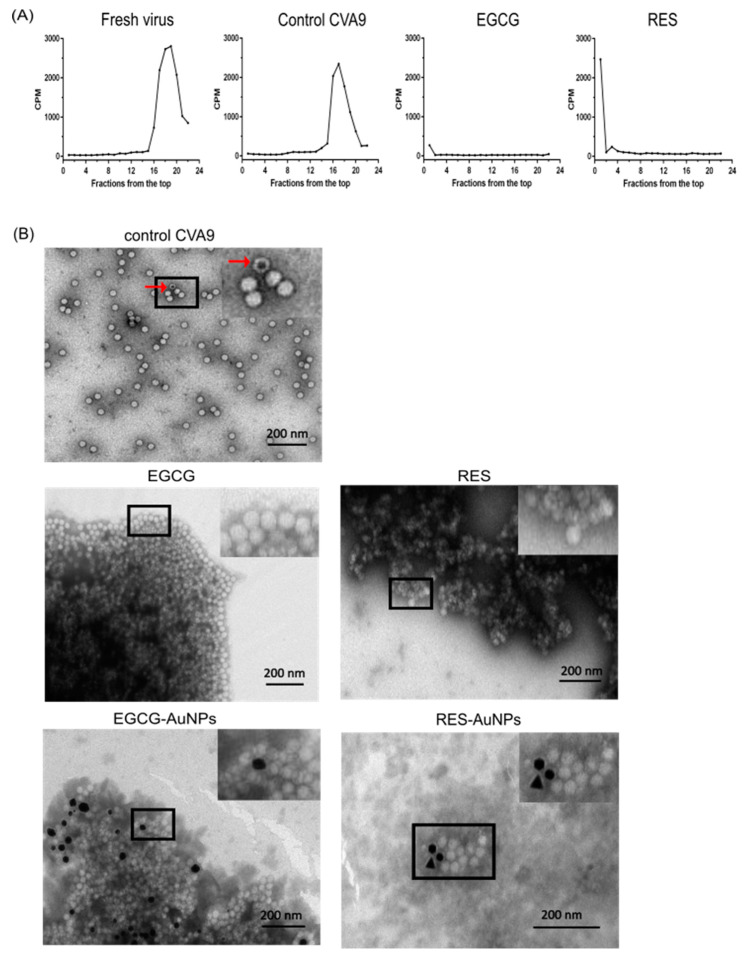
Aggregation/clustering of enteroviruses identified using (**A**) sucrose gradient (5–20%) separation of the metabolically radiolabeled CVA9 after incubation with the polyphenols for 1 h at 37 °C, and (**B**) negatively stained TEM images of CVA9 (1.6 × 10^10^ PFU/mL) and CVA9 treated with EGCG (650 μM), RES (880 μM), EGCG-AuNPs (ligand concentration 5.61 × 10^−1^ μM) and RES-AuNPs (ligand concentration 3.44 μM). Scale bar, 200 nm. The viral particles with a dark inner center (marked with a red arrow in the control CVA9 TEM image) can be easily differentiated as empty/broken virus capsids. (**C**) Dynamic light scattering analysis of EGCG (6.5 μM) with different dilutions of CVA9.

**Figure 6 pharmaceutics-13-01182-f006:**
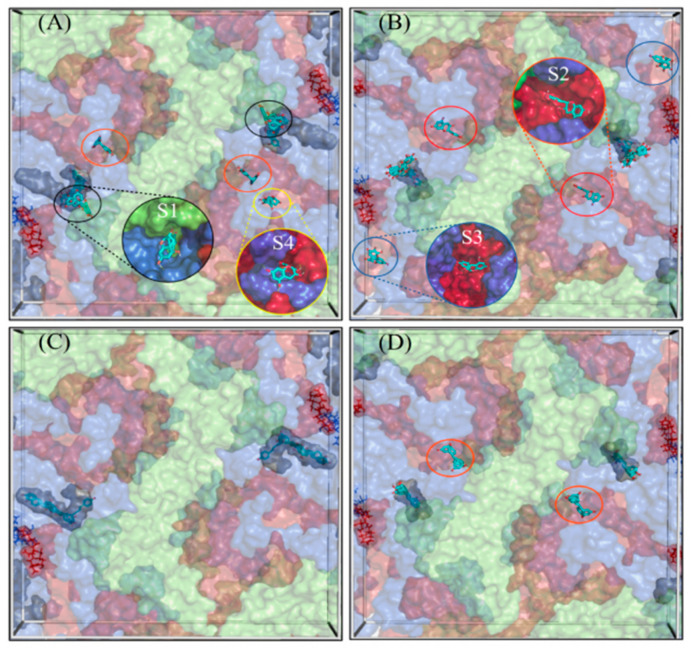
Docking grid boxes centered at the 2-fold axis, for EGCG (top subfigures) and RES (bottom subfigures). CVA9 is on the left (**A**,**C**) while CVB3 is on the right (**B**,**D**). The hydrophobic pocket is represented as a grey surface, while the residues for the Butcher-Neyts pocket are represented as lines. Capsid proteins VP1–VP3 are colored blue, light green and red, respectively. VP4 is not shown. Docked poses are represented as sticks. The binding sites S1, S2 and S3 are colored black, red and blue. Tentative site S4 is in circled in yellow. The additional binding site (S4) was found using EGCG (Figure 1A, circled in yellow) in CVA9 only. There is less statistical support for this site, as only one ligand pose was found in this region. However, with a favorable binding energy (−7 kcal/mol) and an estimated six hydrogen bonds, it cannot be completely ignored. The binding surface for the potential S4 site comprises VP1 residues T274, R275, K276, T280, V281, T282, T283 and V284, and VP3 residues N57, Q59, R86, D92, S93 and V94. As these residues are recorded only from one observation, it is not a consensus for S4.

**Figure 7 pharmaceutics-13-01182-f007:**
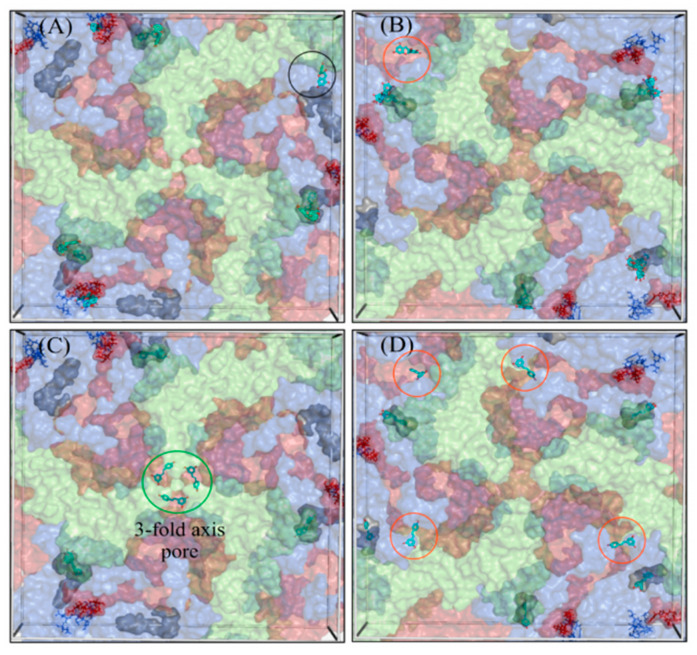
Docking grid boxes centered at the 2-fold axis, for EGCG (top subfigures) and RES (bottom subfigures). CVA9 is on the left (**A**,**C**) while CVB3 is on the right (**B**,**D**). The hydrophobic pocket is represented as a grey surface, while the residues for the Butcher-Neyts pocket are represented as lines. Capsid proteins VP1–VP3 are colored blue, light green and red, respectively. VP4 is not shown. Docked poses are represented as sticks. The binding sites S1, S2 and S3 are colored black, red and blue. Tentative site S4 is in circled in yellow. The additional binding site (S4) was found using EGCG (Figure 1A, circled in yellow) in CVA9 only. There is less statistical support for this site, as only one ligand pose was found in this region. However, with a favorable binding energy (−7 kcal/mol) and an estimated six hydrogen bonds, it cannot be completely ignored. The binding surface for the potential S4 site comprises VP1 residues T274, R275, K276, T280, V281, T282, T283 and V284, and VP3 residues N57, Q59, R86, D92, S93 and V94. As these residues are recorded only from one observation, it is not a consensus for S4.

**Figure 8 pharmaceutics-13-01182-f008:**
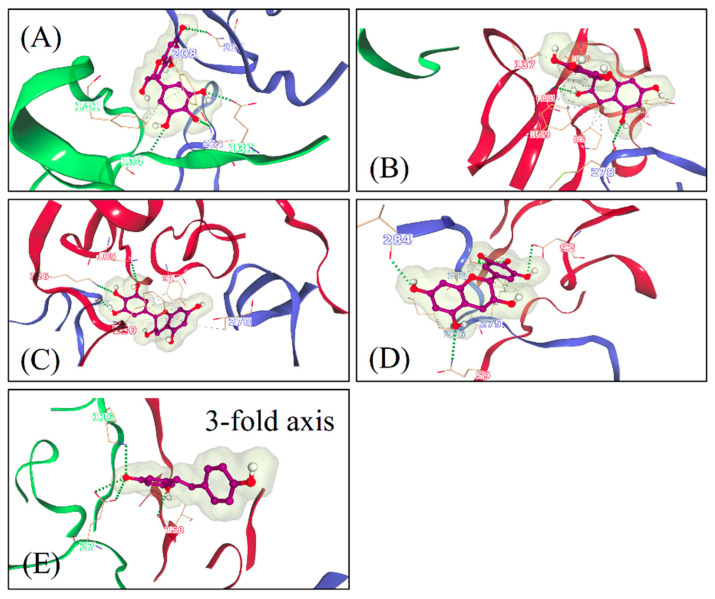
Protein–ligand interactions for example poses obtained at the newly determined binding sites, namely for EGCG at sites (**A**) S1 in CAV9, (**B**) S2 in CBV3, (**C**) S3 in CBV3, (**D**) S4 in CAV9 and (**E**) RES at the 3-fold axis pore in CAV9. Hydrogen bonds and hydrophobic interactions are depicted as dashed green and grey lines, respectively. Interacting residues are labeled, and the capsid proteins VP1–VP3 are colored blue, light green and red, respectively.

**Figure 9 pharmaceutics-13-01182-f009:**
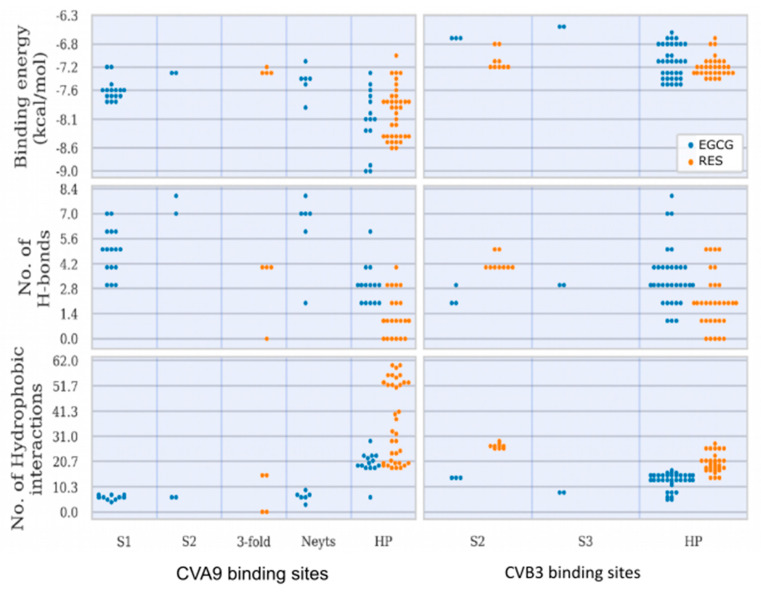
Summary of the binding characteristics of EGCG and RES at each of the six binding sites across CVA9 and CVB3. Each dot is an experimental unit, for which the binding energy, number of H-bonds and number of hydrophobic interactions are shown for CVA9 (left subfigures) and CVB3 (right subfigures). HP—hydrophobic pocket, Neyts—the Butcher-Neyts pocket.

**Table 1 pharmaceutics-13-01182-t001:** Parameters for grid placement along the fold capsid fold axes.

Strain	Fold Axis	Centroid (Å)	Grid Size (in Grid Points)	Shift along x Axis (Å)
CVA9	2	−75.446, 105.465, −13.389	40 × 100 × 100	10
	3	−103.647, 73.405, 5.830	40 × 120 × 120	10
	5	−44.878, 108.985, 50.228	90 × 180 × 180	10
CVB3	2	118.389, 223.579, 288.589	40 × 100 × 100	10
	3	110.197, 256.603, 256.603	40 × 120 × 120	10
	5	183.400, 251.025, 292.820	90 × 180 × 180	10

**Table 2 pharmaceutics-13-01182-t002:** Antiviral activity and cytotoxicity of polyphenols and polyphenol-functionalized AuNPs.

Virus	Compounds	EC_50_ (µM)	Fold Improvement	CC_50_ (µM)	SI
CVB3	NP-EGCG	0.051	67	>1.122	>22.000
	EGCG	3.411		2218.196	650.306
	NP-Res	0.004	26,974	>6.880	>1720.000
	Res	107.894		>4400.000	>40.780
CVB1	NP-EGCG	0.237	5	>1.122	>4.734
	EGCG	1.140		2218.196	1945.785
	NP-Res	0.068	2101	>6.880	>101.176
	Res	142.889		>4400.000	>30.793
CVA9	NP-EGCG	0.004	67	>1.122	>280.500
	EGCG	0.266		2218.196	8339.082
	NP-Res	0.039	8471	>6.880	>176.410
	Res	330.369		>4400.000	>13.318

EC_50_—50% effective concentration, CC_50_—50% cytotoxic concentration, SI—selectivity index.

## Data Availability

Not applicable.

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
