# Peer review of "Polyphenols Epigallocatechin Gallate and Resveratrol, and Polyphenol-Functionalized Nanoparticles Prevent Enterovirus Infection through Clustering and Stabilization of the Viruses"

_pharmaceutics, 2021, doi:10.3390/pharmaceutics13081182_

Round 1

Reviewer 1 Report

Manuscript #pharmaceutics-1299462 titled “Polyphenols, Epigallocatechin Gallate and Resveratrol, and Polyphenol-functionalized Nanoparticles Prevent Enterovirus Infection through Clustering and Stabilization of the Viruses” reports the green synthesis of gold nanoparticles using epigallocatechin gallate (EGCG) and resveratrol (RES). The synthesized gold nanoparticles were evaluated for antiviral activity. Several gold nanoparticles were reported in literature as antiviral (e.g. Viruses. 2019 Dec; 11(12): 1111. doi: 10.3390/v11121111). Herein, EGCG and RES were used as reducing agents. In my humble opinion, the current manuscript should be revised to address the following points:

  1. Characterization of the synthesized nanoparticles should be reported properly.
  2. The calculation of RES and EGCG shown in the supplementary materials should be briefly explained! I found that these calculations were based on assumption from DLS versus TEM data rather than real determination. Thus, the presence of the assumed monolayer is not sufficiently proved. In fact, phenolic compounds and polyphenolic compounds can dissolve in sodium hydroxide. I wonder if washing with 0.02M NaOH has removed all RES and EGCG. In addition, when phenolic compounds is used to reduce gold salts, the phenolic compounds itself are oxidized. Chemical nature of organic materials loaded on the nanoparticles (if present) should be verified.
  3. The used in-house Python scripts should be included in the supplementary materials.
  4. Before accepting docking results, the chemical nature of the organic material loaded (if any) should be verified.
  5. Please, give supporting reference(s) to the statement in the introduction (line 67) that binding polyphenols to gold nanoparticles can reduce toxicity.

Author Response

Manuscript #pharmaceutics-1299462 titled “Polyphenols, Epigallocatechin Gallate and Resveratrol, and Polyphenol-functionalized Nanoparticles Prevent Enterovirus Infection through Clustering and Stabilization of the Viruses” reports the green synthesis of gold nanoparticles using epigallocatechin gallate (EGCG) and resveratrol (RES). The synthesized gold nanoparticles were evaluated for antiviral activity. Several gold nanoparticles were reported in literature as antiviral (e.g. Viruses. 2019 Dec; 11(12): 1111. doi: 10.3390/v11121111).

We are grateful to the reviewer for this citation which we overlook and which was now included in the manuscript.

Herein, EGCG and RES were used as reducing agents. In my humble opinion, the current manuscript should be revised to address the following points:

  1. Characterization of the synthesized nanoparticles should be reported properly.
  2. The calculation of RES and EGCG shown in the supplementary materials should be briefly explained!

We have now added a more detailed explanation for the calculation.

I found that these calculations were based on assumption from DLS versus TEM data rather than real determination. Thus, the presence of the assumed monolayer is not sufficiently proved. In fact, phenolic compounds and polyphenolic compounds can dissolve in sodium hydroxide.

Here we disagree with the reviewer. It is well-known that the molecule which is used for the redox reaction to grow the nanoparticle is also forming a quite stable first surface layer on the gold. Gold surfaces without any attached molecules do not exist in solution. That this layer even if made of highly soluble molecules are not easily removed was shown by Mandal et al (10.1007/s00396-010-2343-2) where the citrate from the redox reaction was found also on the surface after incubation and additional coated with polyanions. This experiment confirms also that it is fair to assume that the first layer is made of the intact molecules of the redox agent. This first layer is only substitute but not solved by a molecule with a high binding affinity such as a thiol. This replacement and the stability of the so-called “hard” protein corona was described in great detail (e.g.10.1021/nn202458g).

In order to confirm this we made a new experiment: we measured of the polyol concentration indirectly by determining the amount of bound cysteine replacing the polyphenol by a colorimetric emthod with Ellman-reagent. The measured concentration confirms the presence of only a monolayer and the value within the error range. This information and method has been added in the manuscript.

 I wonder if washing with 0.02M NaOH has removed all RES and EGCG. In addition, when phenolic compounds is used to reduce gold salts, the phenolic compounds itself are oxidized. Chemical nature of organic materials loaded on the nanoparticles (if present) should be verified. (see explanation above) A sentence was added to the text to explain this.

  1. The used in-house Python scripts should be included in the supplementary materials.

The algorithm has been explained in great detail in the Methodology section . The developer of the algorithm, Dr Olivier Sheik Amamuddy (3rd author in this manuscript), is converting it to a tool for the broad use of it by the scientific community. Drs Sheik Amamuddy and Tastan Bishop will submit a separate manuscript for this in the next few months time. Thus, at this point, we will not be able to make the scripts public in the Supplementary Data. We hope that this is acceptable to the Reviewer. 

  1. Before accepting docking results, the chemical nature of the organic material loaded (if any) should be verified.

This is not necessary as the calculation for the docking was made with the free ligand not considering the nanoparticle bound molecule. However, as explained above it is fair to assume that intact polyphenols are attached to the surface.

  1. Please, give supporting reference(s) to the statement in the introduction (line 67) that binding polyphenols to gold nanoparticles can reduce toxicity.

A sentence with an explanation and a citation was added to the text.

Reviewer 2 Report

The manuscript under review is well designed and presented. The results are supported by a number of experiments and a computational study was also carried out. The manuscript may be considered for publication in Pharmaceutics. However, I suggest following minor amendments:

  1. Figure 5 (B): visibility of this image should be improved.
  2. A figure showing non-bond interactions can provide an elaborated view of the interacting residues involved in stabilizing the docked compounds in the binding site.

Author Response

The manuscript under review is well designed and presented. The results are supported by a number of experiments and a computational study was also carried out. The manuscript may be considered for publication in Pharmaceutics. However, I suggest following minor amendments:

We are grateful for the positive comments by the referee.

  1. Figure 5 (B): visibility of this image should be improved.

We have now increased the size of the TEM figures to make the visibility better.

  1. A figure showing non-bond interactions can provide an elaborated view of the interacting residues involved in stabilizing the docked compounds in the binding site.

We thank the referee for the suggestion. This image, new Fig. 8, has been now added with explanatory text also in the results section. The new figure includes single ligand poses docked at each of the newly found sites.

Reviewer 3 Report

In the present manuscript, Reshamwala et al. claim that natural-occurring polyphenols EGCG and Resveratrol, and polyphenol-functionalized Nanoparticles can inhibit enterovirus infection, although with different degrees of efficacy.

Overall, the manuscript presents original data and might be relevant for the the development of innovative antiviral agents. There are, however, several issues that must be addressed by authors in the revised manuscript, as described below.

1) The authors should discuss the physiological relevance of the polyphenol and polyphenol nanoparticles concentrations used for the cell viability and antiviral activity experiments. Can these concentrations be achieved in vivo with a safe toxicological profile? Which drug delivery systems could be used for optimal results? The authors should further elaborate on the translational relevance of their in vitro findings in their Discussion section.

2) The authors should disclose which statistical analysis they performed, for each experiment, as well as the data analysis software that was used. In addition, I found very unusual that the authors did not display all relevant statistical comparison between experimental groups in their graphs.

3) It has been shown that EGCG and other closely related flavonoids can interact with several disease-causing proteins, namely amyloidogenic proteins and redirect their abnormal polymerization into stable, non-reactive, non-toxic aggregates that are more amenable to cells and tissues (PMID: 21740906; PMID: 19861125; PMID: 29124175; PMID: 22253829; PMID: 23670234; PMID: 20385841). The authors should acknowledge these studies in their Discussion section, and speculate on the potential mechanistic similarities between non-specific EGCG/flavonoid-protein binding interactions observed with disease-causing amyloids and the observed EGCG-induced clustering of virions into more stable conformations.

Author Response

In the present manuscript, Reshamwala et al. claim that natural-occurring polyphenols EGCG and Resveratrol, and polyphenol-functionalized Nanoparticles can inhibit enterovirus infection, although with different degrees of efficacy.

Overall, the manuscript presents original data and might be relevant for the development of innovative antiviral agents. There are, however, several issues that must be addressed by authors in the revised manuscript, as described below.

  • The authors should discuss the physiological relevance of the polyphenol and polyphenol nanoparticles concentrations used for the cell viability and antiviral activity experiments. Can these concentrations be achieved in vivo with a safe toxicological profile? Which drug delivery systems could be used for optimal results? The authors should further elaborate on the translational relevance of their in vitro findings in their Discussion section.

We thank the referee for this good suggestion. We have addressed this issue and added text in the discussion section.

2) The authors should disclose which statistical analysis they performed, for each experiment, as well as the data analysis software that was used. In addition, I found very unusual that the authors did not display all relevant statistical comparison between experimental groups in their graphs.

We have now performed more extensive statistical analysis in the figures 1 and 3 and added the asterisks in those bar graphs. We have added also information of the nature of the statistical testing in the material and methods.

  • It has been shown that EGCG and other closely related flavonoids can interact with several disease-causing proteins, namely amyloidogenic proteins and redirect their abnormal polymerization into stable, non-reactive, non-toxic aggregates that are more amenable to cells and tissues (PMID: 21740906; PMID: 19861125; PMID: 29124175; PMID: 22253829; PMID: 23670234; PMID: 20385841). The authors should acknowledge these studies in their Discussion section, and speculate on the potential mechanistic similarities between non-specific EGCG/flavonoid-protein binding interactions observed with disease-causing amyloids and the observed EGCG-induced clustering of virions into more stable conformations.

This is a very interesting analogy and good suggestion. We have added references and text in the discussion section on this.

Round 2

Reviewer 1 Report

The revised manuscript addressed some raised points presenting new data. Meanwhile, the rebuttal to other points might be accepted. In my humble opinion, the current version of the manuscript might be accepted for publication.  

Reviewer 3 Report

The authors have successfully addressed my concerns, thus I recommend the revised version of the manuscript for publication.